# LASS-ODE: When Large Foundation Models Meet Small Unified ODE Representations

## Abstract

Foundation models have transformed language and vision through large-scale attention over discrete tokens, yet progress on continuous-time dynamical signals remains limited. A core challenge is the absence of a natural token-based representation for ODE trajectories, which evolve continuously, span multiple temporal resolutions, and are often partially observed. We introduce the Tokenized ODE Representation (TOR), which maps trajectories into latent tokens governed by local and linear Neural ODEs, leveraging their linearity for efficient scaling. To capture both temporal context and shared structure across systems, we design a hybrid attention architecture that alternates intra-system self-attention, modeling dependencies within each trajectory, and inter-system cross-attention, supported by a Dynamic ODE Hub (DOH) that serves as a shared repository for inter-system knowledge exchange. These components form LASS-ODE (LArge-Scale Small ODE), a foundation model with strong capacity for interpolation, extrapolation, probabilistic inference, and zero-shot generalization across diverse ODE systems.

## 1 Introduction

Foundation models have achieved transformative progress across modalities such as text, vision, time series, and graphs by leveraging large-scale attention-based architectures (Vaswani et al., 2017; Devlin et al., 2019; Brown et al., 2020; Dosovitskiy et al., 2020; Kaplan et al., 2020; Das et al., 2024a; Xiaoming et al., 2025; Liu et al., 2025b; Mao et al., 2024; Liu et al., 2025a). At their core is the attention mechanism, which enables global token interaction and flexible context aggregation, supported by predictable scaling laws with model and data size (Kaplan et al., 2020). Yet a central paradox remains: while foundation models thrive on *discrete tokens*, many scientific and engineering domains are governed by *continuous-time dynamics* described by Ordinary Differential Equations (ODEs). These dynamics underpin simulations, forecasting, digital twins, and control in engineered systems (Vallado, 2001; Rasheed et al., 2019; Li et al., 2025b; Mai et al., 2025), Earth and environmental processes (Lorenz, 2017), and life and health systems (Qian et al., 2021). Bridging the discrete-token paradigm with continuous-time dynamical systems remains an open challenge.

Existing approaches extend Transformers to temporal data by discretizing trajectories into patches and treating them as tokens. Recent time-series foundation models have advanced this strategy by learning local short-term dynamics within each patch and using attention to capture long-range dependencies (Nie et al., 2022; Das et al., 2024a; Ansari et al., 2024a). While effective for regularly sampled signals, these models remain tied to fixed-interval updates and cannot faithfully represent the underlying ODEs. As a result, they lack guarantees of continuity, struggle with irregular or multi-resolution data, and fail to expose the physical semantics of dynamical systems. Techniques such as imputation and positional encoding can partially address irregularity (Wen et al., 2022; Zeng et al., 2023), but they do not resolve the fundamental gap: patch tokenization does not provide a principled way to encode continuous evolution.

Compared with patch-based foundation models, small-scale system-specific approaches can directly model continuous-time dynamics within a single system. Symbolic regression methods infer explicit governing equations from data, producing closed-form ODEs that can be integrated at arbitrary temporal resolutions (Brunton et al., 2016; Li et al., 2022). Physics-informed neural networks embed physical constraints into coordinate-based neural fields, allowing the learned solution to be queried at any time (Cuomo et al., 2022). The Neural ODE family instead learns the underlying vector

field and integrates it in continuous time (Chen et al., 2018b; Rubanova et al., 2019; Kidger et al., 2020b; Li et al., 2025a). These approaches can achieve high accuracy on targeted systems, yet often require case-specific tuning and lack scalability. More importantly, they do not generalize across heterogeneous systems, even though many systems share recurring ODE structures such as energy-conserving Hamiltonian forms (Greydanus et al., 2019), damping and forcing patterns, or oscillatory dynamics (Krishnapriyan et al., 2021; Rackauckas et al., 2020). This tension highlights the need for architectures that preserve the fidelity of continuous-time modeling while supporting transfer across diverse dynamical domains.

To address this gap, we introduce a new philosophy, *LASS (LArge-Scale Small)*, for scalable modeling of ODE-governed systems. In contrast to conventional strategies that focus solely on scaling up large attention architectures, our philosophy emphasizes first *scaling down* to capture local physical information in small, physics-aware tokens, and then *scaling up* with global attention to integrate these tokens across trajectories and systems. Each token preserves physical constraints and provides an efficient representation of local dynamics, often through simple linear forms, while attention layers support flexible long-range interactions. This dual design—small physics-aware representations combined with large-scale token interaction—aims to achieve both fidelity to underlying physical laws and scalability across diverse datasets.

Building on this philosophy, we propose *LASS-ODE*, a scalable architecture that unifies continuous-time modeling with foundation-style generalization. Central to LASS-ODE is the *Tokenized ODE Representation (TOR)*, which encodes state trajectories as piecewise linear ODE flows in the latent space. Unlike patch tokenizers, TOR preserves continuity and interpretable eigenstructure. Moreover, the linear format brings fast ODE integration, suitable for large-scale computation. To facilitate cross-system generalization, we introduce the *Dynamic ODE Hub (DOH)*, a shared, trainable matrix that is dynamically updated from token messages and serves as a hub for inter-system knowledge exchange. Finally, we propose a novel architecture that alternates between intra- and inter-attention, enabling deep, dynamic mutual enhancement between local dynamics and shared global structures.

Our contributions are threefold:

- **Continuous-time tokenization.** We introduce the Tokenized ODE Representation (TOR), which encodes latent flows into piecewise linear tokens with guaranteed continuity, interpretable eigenstructure, and fast computations.

- **Scalable cross-system modeling.** We design LASS-ODE, a foundation model that blends physics-aware tokenization with scalable intra- and inter-attention mechanisms.

- **Empirical validation.** Across chaotic physics, biological networks, and power systems, LASS-ODE achieves consistent improvements over state-of-the-art time-series foundation models and Neural ODE-based methods.

## 2 Preliminaries

**Problem Setup.** We consider families of continuous-time dynamical systems governed by ordinary differential equations (ODEs) or differential-algebraic equations (DAEs). For a given system with state $\boldsymbol{x} \in \mathcal{X}$, where $\mathcal{X}$ denotes the system's state space, its evolution is described by a vector field $f : \mathcal{X} \to \mathcal{X}$ as $\dot{\boldsymbol{x}} = f(\boldsymbol{x})$. Our setting includes sequential state observations $\{\boldsymbol{x}_i(t_j^i)\}_{i,j}$ from multiple systems indexed by $i$, each with distinct state spaces $\{\mathcal{X}_i\}$, potentially varying in dimension, temporal resolution, and observability. For the $i^{th}$ system, the timestamps of all observed points $\mathcal{T}_{\mathrm{obs}}^i := \{t_j^i\}_j$ are also available. Our goal is to build a foundation model that estimates $f$ for each system. The model supports the following downstream tasks:

- **Interpolation:** Reconstruct missing data at arbitrary times within the observed time span.

- **Extrapolation:** Predict system trajectories at future times beyond the observation interval.

- **Probabilistic trajectory inference:** Estimate distributions over trajectories to enable uncertainty-aware predictions under partial observability, noise, etc.

- **Zero-shot inference:** The model reconstructs trajectories for previously unseen systems by conditioning only on a short prefix of observed states, without access to any additional input-output pairs from the same system or parameter fine-tuning (Liu et al., 2025b).

In the following, we introduce the preliminaries for the learning models and omit the system index for notational simplicity, as all models are applied independently to each system in practice.

**Latent ODE for multi-dimensionality, partial-observability, and multi-resolution.** Recall that different ODE systems can be multi-dimensional, partially observed, and sampled at multiple resolutions. We first *replicate certain channels to enforce a fixed input dimensionality across systems*, but this alone does not address the underlying redundancy or missing information. We therefore *construct a latent space* that captures the core dynamical factors to reduce redundancy (tackling multi-dimensionality) and introduce a latent random variable to represent a distribution over trajectories (tackling partial observability). We then evolve this latent state using a Neural ODE to accommodate multi-resolution sampling. Together, these components motivate the use of Latent ODE (Rubanova et al., 2019). Specifically, in latent ODE, an encoder $f_{\text{enc}}$ processes $N$ observations $\{\boldsymbol{x}(t_i)\}_{i=0}^N$ and $\mathcal{T}_{\text{obs}} := \{t_i\}_{i=0}^N$ to infer an initial latent state $\boldsymbol{z}(t_0) \in \mathcal{Z}$ that summarizes past information. The posterior distribution over this state is parameterized as:

$$\boldsymbol{z}(t_0) \sim \mathcal{N}\big(f_{\text{enc}}(\{\boldsymbol{x}(t_i)\}, \mathcal{T}_{\text{obs}})\big). \tag{1}$$

**Choice of the encoder $f_{\text{enc}}$ for large-scale training.** We adopt a GRU encoder (the orange box in the left of Fig. 1) that processes the concatenated input $[\boldsymbol{x}(t), \Delta t]$, where $\Delta t$ denotes the sampling interval of the specific ODE system. This design enables the encoder to handle multi-resolution datasets by incorporating temporal resolution as an input. Specifically, we have:

$$f_{\text{enc}}(\{\boldsymbol{x}(t_i), \Delta t_i\}_{i=0}^N) = g(\boldsymbol{h}) \,, \boldsymbol{h} = \text{GRU}_\phi(\{[\boldsymbol{x}(t_i); \Delta t_i]\}_{i=0}^N), \tag{2}$$

where $g$ is a multilayer perceptron (MLP) to convert $\boldsymbol{h}$ into the mean and variance of $\boldsymbol{z}(t_0)$. Other options for irregularly sampled data, such as RNN-$\Delta_t$ (Che et al., 2018) and ODE-RNN (Rubanova et al., 2019), incur heavier computational costs and prevent the use of highly optimized batched GRU kernels, which leads to significantly lower GPU throughput.

Given $\boldsymbol{z}(t_0)$, the decoder evolves the latent trajectory via an ODE parameterized by a neural network $h_\theta$, and reconstructs/predicts the observed states using a likelihood model $p_\theta$:

$$\boldsymbol{z}(t_i) = \text{ODESolve}(h_\theta, \boldsymbol{z}(t_0), t_0, t_i), \quad \boldsymbol{x}(t_i) \sim p_\theta(\boldsymbol{x}(t_i) \mid \boldsymbol{z}(t_i)), \tag{3}$$

Here, $\text{ODESolve}(h_\theta, \boldsymbol{z}(t_0), t_0, t_i)$ solves the initial value problem (IVP) defined by the neural ODE $\dot{\boldsymbol{z}}(t) = h_\theta(\boldsymbol{z}(t))$ with initial condition $\boldsymbol{z}(t_0)$, and returns the integrated latent state at time $t_i$. The decoder thus consists of both the ODE function $h_\theta$ and the probabilistic head $p_\theta$. Training maximizes the variational lower bound:

$$\text{ELBO} = \mathbb{E}_{\boldsymbol{z}(t_0)}\left[\log p_\theta(\{\boldsymbol{x}(t_i)\})\right] - \text{KL}\left[\mathcal{N}\big(f_{\text{enc}}(\{\boldsymbol{x}(t_i)\}, \mathcal{T}_{\text{obs}})\big) \,\|\, p(\boldsymbol{z}(t_0))\right], \tag{4}$$

where $p(\boldsymbol{z}(t_0))$ is typically a standard Gaussian prior. In deterministic settings, the likelihood term can be replaced with a squared-error loss.

**Multi-head attention for scalable interactions**. As the core computation underlying modern foundation models, multi-head attention enables the contextual interaction between embedding vectors. Given query $Q$, key $K$, and value $V$, the attention output is computed as:

$$\text{MHA}(Q, K, V) = \text{Concat}(\text{head}_1, \ldots, \text{head}_h)W^O, \quad \text{head}_i = \text{softmax}\left(\frac{QW_i^Q(KW_i^K)^\top}{\sqrt{d_k}}\right)VW_i^V, \tag{5}$$

where $d_k$ is the dimension of each query/key vector.

## 3 LASS-ODE: LARGE-SCALE SMALL ODE

### 3.1 TOKENIZED ODE REPRESENTATION TO SCALE DOWN LATENT ODEs

Latent ODEs provide a principled way to model continuous-time dynamics, but they remain computationally expensive due to the need for numerical ODE integration at every forward pass. Moreover, the neural network $h_\theta$ that parameterizes the latent derivative in equation 3 is typically unstructured and difficult to interpret, which limits its scalability and generalizability across systems. In foundation models, tokens are the essential elements that support large-scale representation learning and

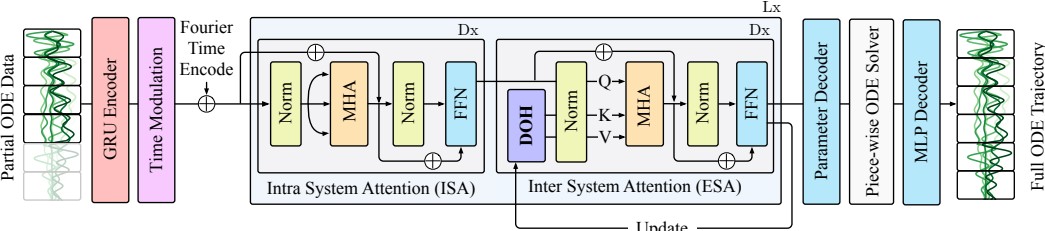

Figure 1: The architecture of LASS-ODE.

compositional reasoning. Hence, before scaling up latent ODEs to the foundation scale, we first scale them down by converting continuous latent dynamics into a small set of discrete ODE tokens. This Tokenized ODE Representation (TOR) reduces integration overhead and imposes structural regularity on each token's dynamics, making them both interpretable and reusable as fundamental units for large-scale modeling.

Given the observations $\{\boldsymbol{x}(t_i)\}_{i=0}^{N}$, we define that the $k^{th}$ token is a small temporal segment of the observed trajectory over the interval $[t_k^{\text{start}}, t_k^{\text{end}}]$. The set of tokens forms a collection of non-overlapping patches of the full time-series trajectory (Nie et al., 2022) from $t_0$ to $t_N$. We now formally define the Tokenized ODE Representation (TOR), which models each trajectory patch using a token-specific latent ODE flow.

**Definition 1** (Tokenized ODE Representation (TOR)). Given tokenized segments, TOR models each token's latent evolution in a continuous space $\mathcal{Z}$ as:

$$\text{TOR}_k := \{\boldsymbol{z}_k(t) \mid t \in [t_k^{\text{start}}, t_k^{\text{end}}]\}, \quad \text{where} \quad \boldsymbol{z}_k(t) = \text{ODESolve}(\dot{\boldsymbol{z}}_k(t), \boldsymbol{z}_k(t_k^{\text{start}}), t_k^{\text{start}}, t), \quad (6)$$

with the token-specific linear dynamics defined by:

$$\dot{\boldsymbol{z}}_k(t) = A_k \boldsymbol{z}_k(t) + \boldsymbol{b}_k, \tag{7}$$

where $\boldsymbol{z}_k(t_k^{\text{start}}) \in \mathcal{Z}$ is the initial latent state, and $A_k \in \mathbb{R}^{d \times d}$, $\boldsymbol{b}_k \in \mathbb{R}^d$ are token-specific parameters. The latent segment $\text{TOR}_k$ can be decoded via $p_\theta(\text{Token}_k \mid \text{TOR}_k)$ to accurately reconstruct the original observation patch, where $p_\theta$ and the decoding process is defined in equation 3.

The linear latent dynamics in equation 7 provide sufficient expressiveness over small token intervals, particularly when composed with the nonlinear map $p_\theta$. TOR retains the key advantages of latent ODEs: it provides a unified representation that naturally supports multi-resolution sampling, continuous evolution, and principled uncertainty modeling through the variational latent variable framework. In addition, because of the linearity, TOR is highly efficient for large-scale computation. During decoding, the initial latent state $\boldsymbol{z}(t_0)$ is encoded once using a GRU-based encoder, as depicted in equation 1 and equation 2. Then, with the shared $\boldsymbol{z}(t_0)$, off-the-shelf ODE solvers such as `torchdiffeq.odeint` (Chen et al., 2018b) can efficiently solve the following piecewise linear IVP, which guarantees the continuity of the ODE trajectories.

$$h(t) := A(t)\boldsymbol{z}(t) + \boldsymbol{b}(t), \quad \text{where} \ A(t) := A_k, \ \boldsymbol{b}(t) := \boldsymbol{b}_k \quad \text{if} \ t \in [t_k^{\text{start}}, t_k^{\text{end}}],$$
$$\boldsymbol{z}(t) = \text{ODESolve}\left(h(t), \ \boldsymbol{z}(t_0), t_0, t \in (t_0, t_{N_{\max}}]\right), \tag{8}$$

where we define the piecewise linear function $h(t)$ to take place of the nonlinear ODE function $h_\theta$ in equation 3. Moreover, we utilize $t_{N_{\max}}$ rather than $t_N$ to denote the user-defined maximum ending time across all ODE systems. The white box in the right part of Fig. 1 is the introduced ODE solver. After obtaining the flow $\boldsymbol{z}(t)$, we can utilize an MLP decoder (see the blue box in the right of Fig. 1), namely, $p_\theta$ in equation 3, to draw samples of $\boldsymbol{x}(t)$.

The introduced physical prior in TOR not only facilitates efficient modeling and computation, but also naturally motivates the use of global attention in a foundation model: each token embedding should attend to others to exchange and refine dynamic patterns, such as oscillatory modes and damping behaviors, reflected in the eigenstructure of $A_k$ and the offset $\boldsymbol{b}_k$ (Callier & Desoer, 2012). To support TOR-based foundation models, two key questions naturally arise:

1. How can we construct an informative embedding for each token?

2. How should we design intra- and inter-ODE system attention?

The first question is addressed below, while the second is discussed in Section 3.2.

To represent each token, we summarize its temporal evolution using the hidden state in equation 2. Compared to the latent initial condition $z(t_0)$, the hidden state $h$ provides a richer and higher-dimensional summary, as also noted in Rubanova et al. (2019). To further encode token-specific dynamics such as $A_i$ and $b_i$, we adopt a time modulation, inspired by (Perez et al., 2018):

$$e_i = \gamma(t_i^{\text{start}}) \odot f_{\text{embed}}(h) + \beta(t_i^{\text{start}}), \qquad (9)$$

where $e_i \in \mathbb{R}^{d_{\text{model}}}$ is the embedding vector for token $i$, and $f_{\text{embed}}$ is an MLP applied to the global summary vector $h$. The modulation functions $\gamma(\cdot)$ and $\beta(\cdot)$ are two independent MLPs that take the token's starting time $t_i^{\text{start}}$ as input and output vectors in $\mathbb{R}^{d_{\text{model}}}$. The operator $\odot$ denotes Hadamard multiplication. By modulating the shared representation $f_{\text{embed}}(h)$ with time-specific scaling and shifting functions $\gamma(t_i^{\text{start}})$ and $\beta(t_i^{\text{start}})$, the model can adaptively adjust the embedding for each token to reflect its unique temporal context. The time modulation is presented as the pink box to the left of Fig. 1.

Since the modulation is conditioned only on the token's starting time $t_i^{\text{start}}$, we can generate the full trajectory of embedding vectors, spanning from the beginning $t_0$ to the defined maximum time $t_{N_{\max}}$, which may extend beyond the end of the observational window $t_N$. As long as the local dynamics $A_k$ and $b_k$ can be accurately estimated from the token embedding (and the subsequent attention), the entire trajectory can be reconstructed by equation 8. This design enables full trajectory reconstruction in a sequence-to-sequence (seq-to-seq) manner, eliminating the need for auto-regressive (AR) next-token prediction (Rubanova et al., 2019).

**Remark: Seq-to-seq vs. AR.** AR generation and next-token prediction have proven highly effective in LLMs. However, in the context of latent dynamics, applying AR introduces a significant computational drawback. Specifically, if we predict the next token in an AR fashion and feed it back as input to the GRU encoder, the latent initial condition $z_0$ in equation 1 may change at each step. As a result, the ODE in equation 8 must be re-integrated repeatedly from a new initial condition. In contrast, our seq-to-seq design enables one-pass ODE integration in each forward round.

### 3.2 Intra- and Inter-system Attention to Scale Up TORs

**Intra-system attention**. We begin by modeling intra-system attention over the full trajectory of tokens. Let $K_{\max}$ denote the number of tokens spanning from the initial time $t_0$ to the extrapolated horizon $t_{N_{\max}}$. Let $E \in \mathbb{R}^{K_{\max} \times d_{\text{model}}}$ denote the matrix of token embeddings $\{e_i\}$, constructed via time modulation as described in equation 9. To further enrich temporal representations, we augment each token embedding with a Fourier feature encoding of its starting time, denoted by $\text{FF}(t_i^{\text{start}})$ (Tancik et al., 2020). Let $\text{FF} \in \mathbb{R}^{K_{\max} \times d_{\text{model}}}$ denote the matrix formed by stacking these encodings for all tokens. These Fourier features enable the model to capture both low- and high-frequency temporal patterns, which are essential for modeling complex and potentially oscillatory system dynamics. The resulting embeddings are processed through a multi-head self-attention (MHA) mechanism, as defined in equation 5, where each token attends to all others within the same trajectory to capture intra-system dependencies. To improve training stability, we adopt the pre-normalization (pre-LN) Transformer (Xiong et al., 2020). Formally, the *intra-system attention module* is defined as:

$$E^{(1)} = E + \text{FF} + \text{MHA}(\text{LN}(E + \text{FF}), \text{LN}(E + \text{FF}), \text{LN}(E + \text{FF})), \qquad (10)$$
$$\tilde{E} = E^{(1)} + \text{FFN}(\text{LN}(E^{(1)})), \qquad (11)$$

where LN denotes layer normalization, and the FFN is a position-wise feedforward network. $\tilde{E} \in \mathbb{R}^{K_{\max} \times d_{\text{model}}}$ is the output. In general, the module is denoted as $\tilde{E} = \text{ISA}(E)$.

**Inter-system attention.** Unlike intra-system attention, inter-system interaction lacks an inherent sequential structure. To enable cross-trajectory communication in a permutation-invariant manner, we draw inspiration from Set Attention (Lee et al., 2019) and Slot Attention (Locatello et al., 2020).

Specifically, we introduce a *Dynamic ODE Hub (DOH)* as the central mechanism. The DOH serves two key roles: (1) it interacts with token embeddings from all systems to extract shared structure and dependencies, and (2) it provides a higher-level abstraction that dynamically responds to incoming token information, enabling cross-system coordination.

Let $H \in \mathbb{R}^{S \times d_{\text{model}}}$ denote a trainable parameter matrix (i.e., a hub) for DOH, where $S$ is a hyper-parameter to control the size of $H$. We perform cross-attention from tokens to the DOH: each token embedding queries the hub to extract information. Formally, the inter-system attention module is defined as:

$$E^{(1)} = E + \text{MHA}(\text{LN}(E), \text{LN}(H), \text{LN}(H)), \tag{12}$$

$$\tilde{E} = E^{(1)} + \text{FFN}(\text{LN}(E^{(1)})), \tag{13}$$

where $\text{LN}(\cdot)$ denotes layer normalization, $\text{MHA}(\cdot, \cdot, \cdot)$ is multi-head attention, and $\text{FFN}(\cdot)$ is a position-wise feedforward network. We denote the overall module as: $\tilde{E} = \text{ESA}(E, H)$. While the static $H$ stores useful global patterns, its expressiveness becomes limited as the ISA and ESA modules are stacked deeper. To capture higher-level abstractions, we promote dynamic updates to $H$. Similar to (Locatello et al., 2020), we allow each hub to evolve based on interactions with token embeddings via cross-attention, and update the hub using a GRU mechanism:

$$\tilde{H} = \text{GRU}\left(\text{MHA}(\text{LN}(H), \text{LN}(E), \text{LN}(E)), H\right), \tag{14}$$

where the GRU treats $H$ as the hidden state and updates it using the cross-attention output as input. This cross-attention allows each hub to selectively aggregate information from all token embeddings. We denote this GRU-based update as $\tilde{H} = \text{Update}(H, E)$.

**Alternating ISA, ESA, and Hub Update**. With the intra-system attention module (**ISA**), inter-system attention module (**ESA**), and the dynamic hub update mechanism (**Update**), we construct the fundamental building block of our architecture. The core idea is to alternate between ISA and ESA to enable bidirectional information flow between local temporal structures and global coordination. We then apply a gating mechanism $G = \text{Gate}(E, E_{\text{inter}})$ to adaptively fuse the intra-attended embeddings $E$ with the DOH-modulated context $E_{\text{inter}}$, enabling selective integration. Finally, the DOH is updated via the **Update** operation to reflect higher-level abstractions and to better support subsequent attention layers. This ISA-ESA-Update pattern forms the core iterative structure of our model, as summarized in Algorithm 1. The middle of Fig. 1 visualizes this process.

---

**Algorithm 1** ISA-ESA-Update

---

**Require:** Token embeddings $E \in \mathbb{R}^{K_{\max} \times d_{\text{model}}}$, initial DOH $H \in \mathbb{R}^{S \times d_{\text{model}}}$
**Require:** Number of blocks $L$, number of attention layers per block $D$
 1: **for** block $\ell = 1$ to $L$ **do**
 2:     **for** layer $d = 1$ to $D$ **do**
 3:         $E \leftarrow \text{ISA}(E)$                    ▷ Intra-system self-attention
 4:     **end for**
 5:     **for** layer $d = 1$ to $D$ **do**
 6:         $E_{\text{inter}} \leftarrow \text{ESA}(E, H)$          ▷ Inter-system attention from DOH
 7:     **end for**
 8:     $G \leftarrow \text{Gate}(E, E_{\text{inter}})$         ▷ Gating function over token and hub context
 9:     $E \leftarrow E + G \odot E_{\text{inter}}$                    ▷ Gated fusion
10:     $H \leftarrow \text{Update}(H, E)$                        ▷ Update hub via GRU
11: **end for**
12: **return** Final token features $\tilde{E} \leftarrow E$

---

### 3.3 TRAINING WITH PREFIXES TO SUPPORT ZERO-SHOT INFERENCE

As discussed in the last paragraph of Section 3.1, we abandon the AR formulation due to the in-tractability of repeatedly integrating large-scale ODEs from updated initial conditions. Instead, we adopt a seq-to-seq approach. However, this design choice may compromise the model's ability to generalize in zero-shot settings, where only a limited temporal prefix is available at test time. To address this, we introduce *prefix-based training* to explicitly teach the model to operate under variable-length observation windows.

Concretely, we simulate zero-shot scenarios during training by varying the prefix length of the input trajectory. This affects only the encoder, i.e., the GRU-$\Delta t$ module, which processes the sequence up to the specified prefix endpoint. We extract the hidden state $\boldsymbol{h}$ at the final prefix time as the summary representation. Since the GRU is causal and recurrent, we can compute all prefix representations in a single forward pass, ensuring efficiency. Formally, let $\{t_m^{\text{prefix}}\}_{m=1}^M$ denote a set of prefix times, where each $t_m^{\text{prefix}} \in [t_0, t_N]$. The encoder GRU-$\Delta t$ processes the full input sequence $\{[\boldsymbol{x}(t_i); \Delta t_i]\}_{i=0}^N$, and we collect the hidden states at these prefix times as $\{\boldsymbol{h}_m\}_{m=1}^M$. Correspondingly, the initial latent states for decoding are defined as $\{\boldsymbol{z}^{(m)}(t_0)\}_{m=1}^M$, computed through equation 1 and equation 2.

Given the prefix-specific GRU hidden state $\boldsymbol{h}_m$ and its corresponding latent initialization $\boldsymbol{z}^{(m)}(t_0)$, we generate token embeddings $E$ using the time-modulated encoding defined in equation 9. These embeddings are then refined through intra- and inter-system attention using the core attention block described in Algorithm 1, resulting in the final token representations $\tilde{E}$.

Next, we introduce a parameter decoder $f_{\text{param}}$ that maps each refined token embedding $\tilde{\boldsymbol{e}}_i$ to a local linear dynamic operator $(A_i, \boldsymbol{b}_i)$, representing the parameters of a piecewise latent ODE:

$$(A_i, \boldsymbol{b}_i) = f_{\text{param}}(\tilde{\boldsymbol{e}}_i), \quad \text{for } i = 1, \ldots, K_{\max}. \tag{15}$$

The parameter decoder is shown as the blue box in Fig. 1. The latent trajectory $\boldsymbol{z}^{(m)}(t)$ is then obtained by integrating the piecewise TOR defined in equation 8, generating the full latent evolution over the interval $[t_0, t_{N_{\max}}]$. Finally, the latent trajectory is decoded into the observable space using equation 3, producing the reconstructed trajectory $\boldsymbol{x}^{(m)}(t)$. This process enables *probabilistic inter-polation and extrapolation*, where the model generates full trajectory reconstructions conditioned on latent dynamics. To train the model, we maximize ELBO defined in equation 4, which is evaluated over the complete time range from $t_0$ to $t_{N_{\max}}$.

## 4 EXPERIMENTS

### 4.1 DATASETS

We evaluate LASS-ODE on a curated dataset across diverse real-world ODE systems, as summarized in Figure 2. Representative training trajectories are provided in Appendix A.2.

For power systems, we study transient stability by integrating swing-equation and higher-order generator-network ODEs using `ANDES` for scripted nonlinear dynamics under perturbations (Huang et al., 2021), and `Dynawo` for utility-grade short- and long-term studies (Guironnet et al., 2018). To probe nonlinear control, we adopt benchmark tasks from the `neural_clbf` library (Amos et al., 2022), which emphasize stability and safety under Lyapunov-Barrier constraints. Beyond engineered systems, we source mechanistic models from the `BioModels` repository (Malik-Sheriff et al., 2020), a curated collection of literature-based ODE models in systems biology, and run time-course simulations for reproducibility. We also evaluate on the "Datasets for learning of unknown characteristics of dynamical systems" (Szczęsna et al., 2023), which comprises 33,000 standardized time series from 15 chaotic and non-chaotic systems with randomized initial conditions. Finally, we include general nonlinear signals from (Augustyn, 2020), covering diverse dynamical systems with partially known or unknown governing equations.

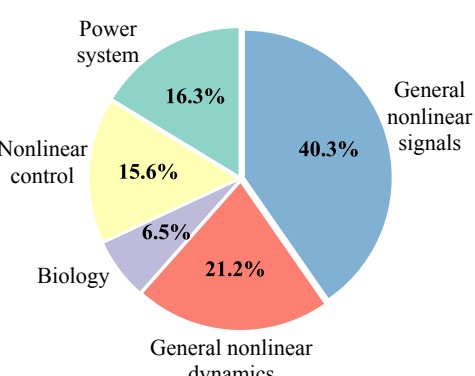

Figure 2: Ratios of data sources in our collected ODE benchmark.

## 4.2 IN-DOMAIN INTERPOLATION AND EXTRAPOLATION

To evaluate in-domain inference, we construct test prefixes from ODE trajectories generated using the same underlying equations as the training data, but with different initial conditions and parameter values. Each prefix randomly spans between $10\%$ and $90\%$ of the full trajectory duration, covering a subinterval of $[t_0, t_{N_{\max}}]$. The computed average Mean Square Error (MSE) is reported in Table 1. We compare our method with both foundation time-series models (TimeFM (Das et al., 2024a), TimeGPT (Garza et al., 2023a), and Chronos (Ansari et al., 2024a)) and Neural ODE-based small-scale methods (Latent ODE (Rubanova et al., 2019) and ContiFomer (Chen et al., 2023)).

Across diverse datasets, LASS-ODE achieves a 50–90% reduction in error compared to competing methods. Firstly, we observe that time-series foundation models are not suitable for inferring ODE measurements since ODE trajectories often exhibit non-periodic behaviors, sharp spikes, oscillatory trends, and irregular patterns (see Appendix A.2 and A.3) that cannot be captured by fixed patch-based tokenization or discrete updates. Secondly, Neural ODE-based methods achieve relatively strong performance within individual systems, highlighting the potential of latent spaces with continuous ODE flows, including our proposed TOR. However, strong in-domain accuracy does not necessarily translate to inter-system generalization. In the next subsection, we therefore evaluate the zero-shot generalization capacity of all methods.

Table 1: Test MSE ($\times 10^{-2}$) for in-domain systems.

| Data | TimesFM Interp | TimesFM Extra | TimeGPT Interp | TimeGPT Extra | Chronos Interp | Chronos Extra | ContiFormer Interp | ContiFormer Extra | Latent ODE Interp | Latent ODE Extra | LASS-ODE Interp | LASS-ODE Extra |
|---|---|---|---|---|---|---|---|---|---|---|---|---|
| Synthetic Nonlinear ODE | - | 0.74 | - | 0.72 | - | 0.90 | 0.48 | 0.62 | 0.38 | 0.19 | **0.08** | **0.09** |
| Power System Swing Equation | - | 7.45 | - | 6.70 | - | 8.94 | 1.45 | 1.63 | 1.81 | 1.90 | **0.67** | **0.74** |
| Linear Fading System | - | 1.49 | - | 1.24 | - | 2.23 | 1.46 | 1.62 | 1.13 | 1.14 | **0.11** | **0.12** |
| Harmonic Oscillator | - | 1.93 | - | 1.55 | - | 2.26 | 0.82 | 0.34 | 0.45 | 0.36 | **0.12** | **0.14** |
| Ueda Oscillator | - | 2.53 | - | 1.92 | - | 2.88 | 0.91 | 0.47 | 0.71 | 0.43 | **0.14** | **0.16** |
| Lorenz Attractor | - | 5.44 | - | 3.96 | - | 6.41 | 0.96 | 0.38 | 0.53 | 0.67 | **0.27** | **0.30** |
| Rucklidge Attractor | - | 8.02 | - | 5.39 | - | 9.67 | 0.63 | 1.17 | 0.82 | 0.86 | **0.35** | **0.38** |
| Rössler Attractor | - | 15.24 | - | 7.62 | - | 19.05 | 1.85 | 1.74 | 1.69 | 1.77 | **0.56** | **0.63** |

## 4.3 ZERO-SHOT GENERALIZATION TO UNSEEN SYSTEMS

Table 2: Test MSE ($\times 10^{-2}$) for unseen systems.

| Data | TimesFM Interp | TimesFM Extra | TimeGPT Interp | TimeGPT Extra | Chronos Interp | Chronos Extra | ContiFormer Interp | ContiFormer Extra | Latent ODE Interp | Latent ODE Extra | LASS-ODE Interp | LASS-ODE Extra |
|---|---|---|---|---|---|---|---|---|---|---|---|---|
| Spiral | - | 0.38 | - | 0.17 | - | 0.29 | 0.35 | 0.47 | 0.17 | 0.16 | **0.03** | **0.05** |
| Glycolytic | - | 15.77 | - | 23.34 | - | 22.79 | 36.39 | 46.98 | 24.15 | 26.90 | **2.35** | **4.08** |
| Lotka | - | 13.37 | - | 17.74 | - | 13.87 | 14.09 | 16.28 | 13.89 | 14.97 | **2.78** | **3.65** |
| SIR Epidemic Model | - | 2.67 | - | 2.14 | - | 2.77 | 3.55 | 3.98 | 3.08 | 3.17 | **0.77** | **1.06** |
| Double Pendulum | - | 57.98 | - | 55.09 | - | 62.89 | 97.82 | 95.69 | 74.63 | 82.23 | **8.23** | **10.22** |
| FitzHugh–Nagumo Model | - | 23.96 | - | 25.55 | - | 28.63 | 31.77 | 35.25 | 20.03 | 18.65 | **5.82** | **7.63** |

To evaluate zero-shot generalization, we use the first 30% of each trajectory as input for interpolation and extrapolation tasks. Results are reported in Table 2. Under this setting, ContiFormer and Latent ODE perform poorly on several complex systems, such as Glycolytic dynamics, the double pendulum, and the FitzHugh–Nagumo model. This limitation arises from their lack of scale-down capacity: they process trajectories sequentially with fixed patterns, making it difficult to adapt when trajectory segments differ across systems. In contrast, LASS-ODE decomposes trajectories into TORs and leverages attention-based interactions between them. This design allows LASS-ODE to maintain low MSE even in few-shot scenarios.

## 4.4 ABLATION STUDY

To assess the contribution of each component, we conduct an ablation study. We test the zero-shot results for different systems and compute the average MSE. As shown in Fig. 3, inter-attention plays a central role in enabling information exchange across different ODE systems, while intra-attention is essential for token contextualization within a trajectory. In addition, removing the time modulation layer leads to a notable increase in MSE. This is because that time modulation in equation 9 creates proper embedding by effectively integrating the temporal feature $h$ with time.

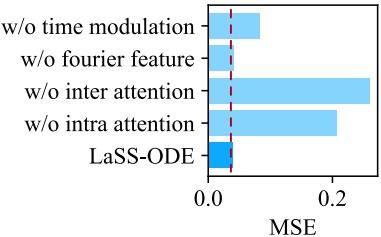

Figure 3: Results in ablation studies.

## 5 RELATED WORK

**Foundation models for time series.** Large-scale pretraining has recently enabled time-series models with cross-domain generalization. TIMEGPT introduced a GPT-style decoder for zero-shot forecasting across diverse corpora (Garza et al., 2023b), and LAG-LLAMA scaled decoder-only transformers with lagged covariates for probabilistic prediction (Rasul et al., 2023). MOIRAI unified training across frequencies and variables with any-variate attention (Liu et al., 2024), while sparse MoE extensions reduce reliance on handcrafted frequency heuristics. Tokenization-based approaches such as CHRONOS and TIMESFM leverage scaling-and-quantization or flexible horizon objectives to achieve strong zero-shot accuracy across benchmarks (Ansari et al., 2024b; Das et al., 2024b). Collectively, these models demonstrate the promise of foundation-style architectures for time series, but they almost exclusively cast sequences into symbolic tokens, overlooking domain-specific inductive biases such as ODE/DAE structure. Our work addresses this gap by developing a foundation-style model that explicitly incorporates continuous-time ODE dynamics and enable transfer across heterogeneous dynamical systems.

**Neural ODE/CDE/SDE families for learning dynamics.** Continuous-time neural architectures offer a complementary line of work. Neural ODEs (Chen et al., 2018a) introduced black-box differential solvers as network layers, enabling flexible vector-field learning. Extensions such as Latent ODEs and ODE-RNNs (Rubanova et al., 2019) infer latent initial states from irregular observations; Neural CDEs generalize recurrent networks to pathwise control signals (Kidger et al., 2020a); and Neural RDEs extend to rough paths (Morrill et al., 2021). These families establish the feasibility of learning dynamics directly in continuous time, but they are typically trained as narrow, system-specific models without cross-domain scalability. Our contribution is to integrate this expressivity into a foundation model for ODEs/DAEs, combining universal pretraining and retrieval with continuous-time architectures to achieve systematic generalization across dynamical regimes under partial observability and noise.

## 6 CONCLUSION, LIMITATION, AND FUTURE WORK

We introduced LASS-ODE, a new foundation model paradigm that unifies large-scale attention with small-scale ODE representations. By tokenizing continuous-time trajectories into piecewise linear latent ODEs, our Tokenized ODE Representation (TOR) enables efficient, interpretable, and reusable modeling units. Together with intra-system self-attention and inter-system cross-attention via a Dynamic ODE Hub, LASS-ODE achieves strong performance across interpolation, extrapolation, probabilistic inference, and zero-shot generalization tasks. Experiments on diverse ODE and DAE datasets demonstrate that our approach bridges the gap between task-specific continuous-time models and general-purpose foundation models. Despite these advances, our work has limitations. Currently, we focus on autonomous systems. On the application side, extending LASS-ODE to digital twins, control-theoretic tasks, and multi-modal settings (e.g., coupling ODE dynamics with text or sensor metadata) offers promising avenues. More broadly, we envision LASS-ODE as a stepping stone toward foundation models that combine the universality of large-scale pretraining with the fidelity of continuous-time physical modeling. For example, our next step is to provide products of LASS-SDE or LASS-PDE.

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

# A APPENDIX

## A.1 MODEL HYPERPARAMETERS AND LEARNABLE PARAMETERS

**Hyperparameters (default run).** Unless otherwise stated, our experiments use the following configuration:

Table 3: Model and training hyperparameters.

| Category | Name | Value |
|---|---|---|
| Architecture | Latent dim | 20 |
| | Embed dim | 256 |
| | GRU hidden | 256 |
| | GRU layers | 2 |
| | Tokens per horizon | 50 |
| | Primitives | 30 |
| | Attention heads | 8 |
| | Attention depth (per stack) | 10 |
| | MLP ratio (FFN) | 4 |
| | Blocks (interleaved) | 2 |
| | ODE solver | `dopri5` |
| Training | Optimizer | AdamW |
| | Learning rate | $5 \times 10^{-4}$ |
| | Betas | $(0.9, \ 0.95)$ |
| | Weight decay | 0.05 |
| | Batch size | 32 |
| | Epochs | 200 |
| | # Scenarios | 15 |
| | Time points | 100-400 |

**Parameter count (by component).** For the default configuration above (and a small output dimension placeholder $D=8$ for the reconstruction MLP), the learnable parameter counts are:

| Module | # Params | Share |
|---|---|---|
| Encoder (GRU-$\Delta t$) | 372,260 | 1.98% |
| Time modulation embedder | 381,440 | 2.02% |
| Fourier positional encoding | 0 | 0.00% |
| *Intra* token self-attention stack (depth 10) | 9,871,360 | 52.38% |
| *Inter* token-primitive cross-attn stack (depth 10) | 7,902,720 | 41.93% |
| Parameter decoder (to $A, b$) | 173,732 | 0.92% |
| Reconstruction MLP ($d \rightarrow D$) | 3,760 | 0.02% |
| **Total (LASS-ODE)** | **18,845,304** | **100%** |

*Notes.* (i) The total is dominated by the attention stacks; changing $E$, heads, or depth has the largest effect. (ii) The reconstruction MLP depends on $D$ (data dimension). If $D$ differs from 8, only that row and the total change slightly. (iii) The primitive library $H \in \mathbb{R}^{K \times E}$ is learnable and contributes linearly in $K \cdot E$.

## A.2 EXAMPLES OF TRAINING TRAJECTORIES

We present representative trajectories from three dynamical systems used in our experiments. These examples highlight the diversity of behaviors in the dataset, spanning chaotic attractors, oscillatory dynamics, and converging swing equations, and provide context for the types of signals on which LASS-ODE is trained.

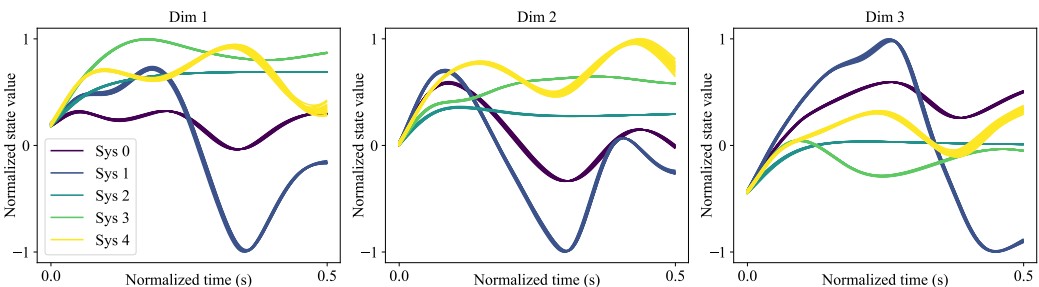

Figure 4: Dataset of general nonlinear signals (Augustyn, 2020).

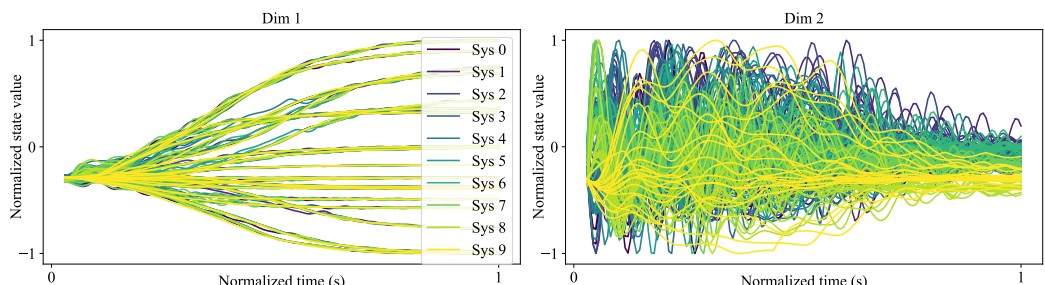

Figure 5: Dataset of power system dynamics governed by swing equations (Huang et al., 2021).

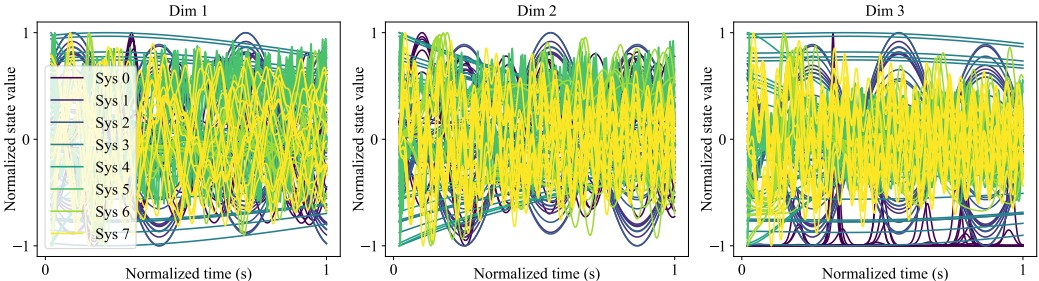

Figure 6: Dataset of general nonlinear dynamics Szczęsna et al. (2023), comprising canonical examples such as the Lorenz attractor, Ueda oscillator, etc.

## A.3 EXAMPLES OF FORECASTING RESULTS

To complement the training trajectories shown in Appendix A.2, we present forecasting results for the same three dynamical systems. These examples demonstrate how LASS-ODE captures distinct dynamical behaviors ranging from chaotic to oscillatory and convergent regimes, highlighting its effectiveness across diverse nonlinear ODE systems.

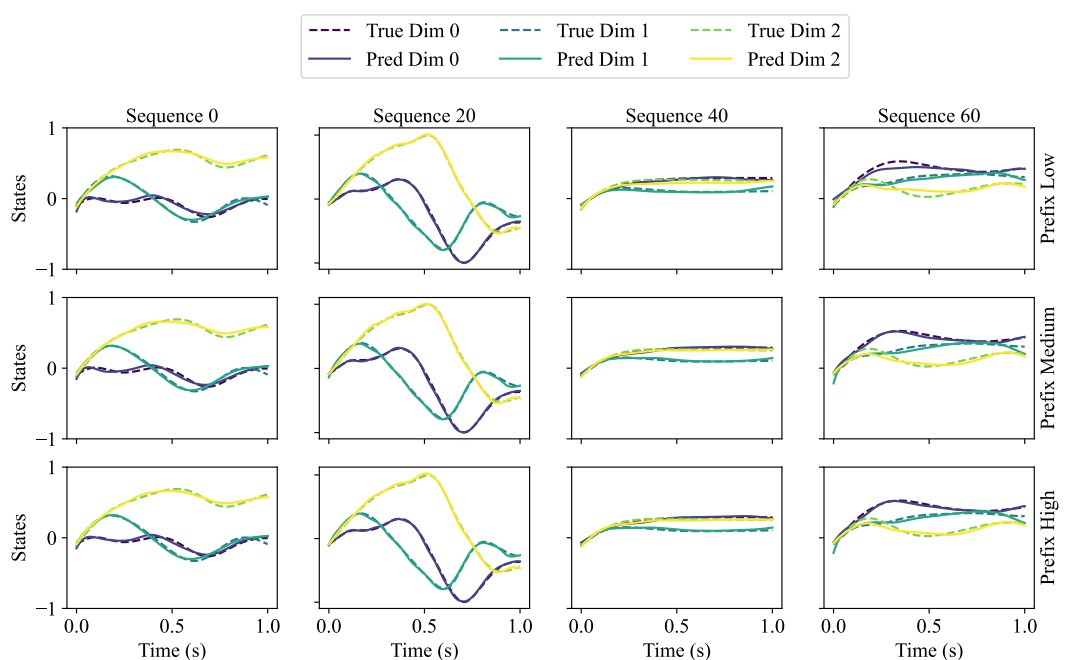

Figure 7: Forecasting results on general nonlinear signals (Augustyn, 2020).

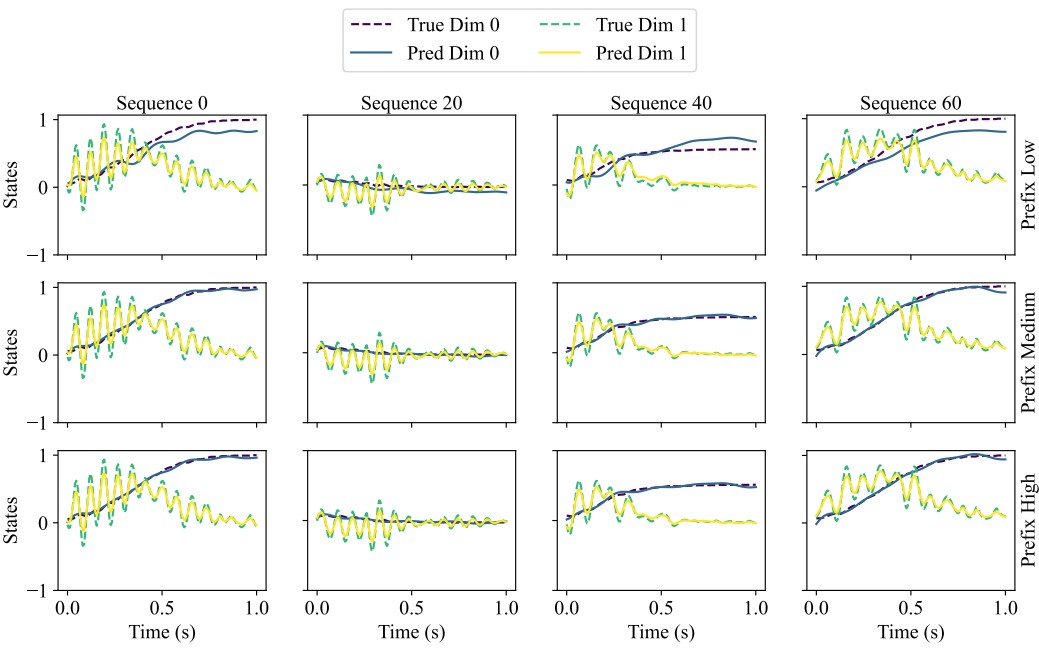

Figure 8: Forecasting results on swing-equation dynamics (Huang et al., 2021), demonstrating accuracy in convergent regimes of power systems.

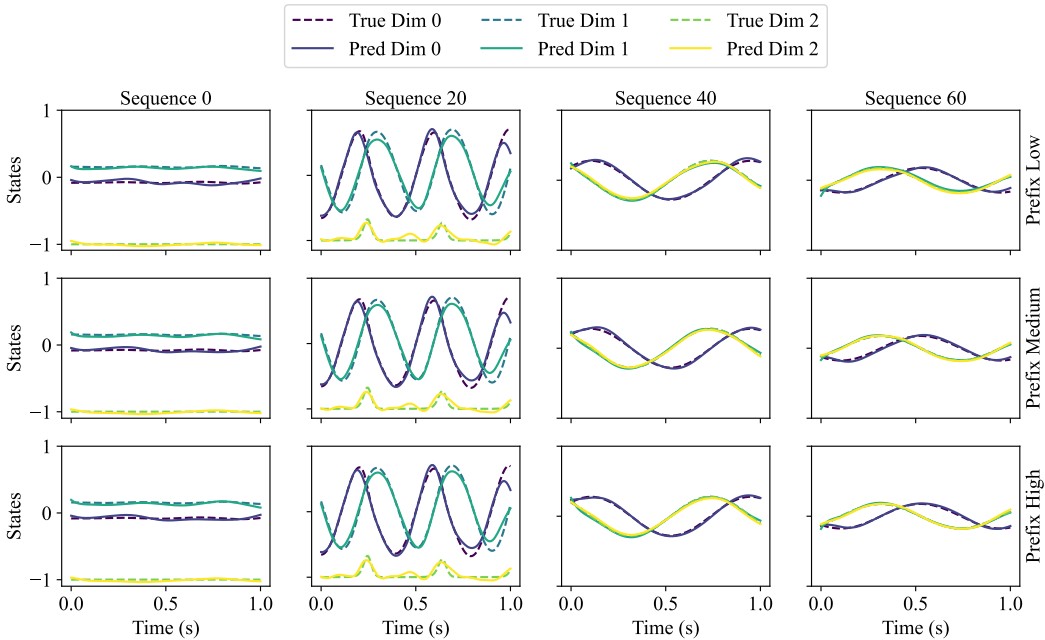

Figure 9: Forecasting results on general nonlinear dynamics (e.g., Lorenz attractor and Ueda oscillator) (Szczęsna et al., 2023; Augustyn, 2020).

## A.4 FORECASTING COMPARISON ON SWING-EQUATION DATASET

To further evaluate forecasting performance, we compare LASS-ODE against recent time-series models, including TimeGPT, TimeFM, and Chronos, on the swing-equation dataset. The models are trained on system trajectories from 0–0.5 s and evaluated on the forecasting horizon 0.5–1 s. This comparison highlights the advantages of physics-informed modeling in capturing power system dynamics relative to general-purpose forecasting baselines.

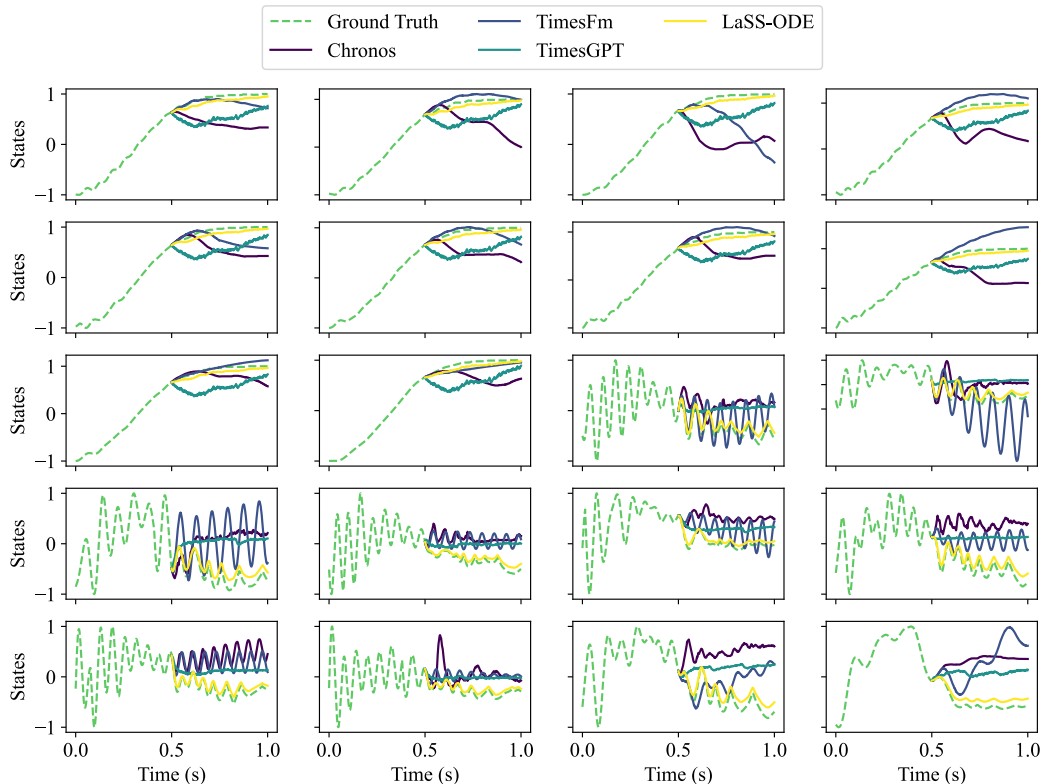

Figure 10: Forecasting comparison on the swing-equation dataset of 20 representative dimensions. Each subplot corresponds to one dimension, with ground-truth trajectories shown as dashed lines, and forecasts from LASS-ODE, TimeGPT, TimeFM, and Chronos shown as solid curves.

