# OpenReview forum: "LASS-ODE: When Large Foundation Models Meet Small Unified ODE Representations"
_ICLR.cc/2026/Conference — ICLR 2026 Conference Withdrawn Submission_

### Official Review · Reviewer_HxGY · 2025-10-29

**Soundness:** 2
**Presentation:** 1
**Contribution:** 2
**Rating:** 2
**Confidence:** 4

**Summary:**

This paper proposes a novel approach for zero-shot time series models that interpolate and extrapolate in continuous time, using multiple sequences as context.

The authors extend the common patching approach by deriving a tokenized ODE representation per patch.  After sharing information between patches and sequences, these tokens estimate a local linear neural ODE per patch in latent space. Solving the resulting piece-wise linear ODE and projecting it to data space yields a continuous-time model prediction.

The authors evaluate their method against existing zero-shot time series forecasting models, LatentODE and ContiFormer. Evaluation is performed on synthetic data of systems modeling real world phenomena.

**Strengths:**

1. The authors tackle an important issue: extending the existing zero-shot time series paradigm to continuous time (e.g. to better handle irregular grid data).
2. The overall idea of sharing information between patches and learning local neural ODEs is interesting.
3. They provide a small but insightful ablation study on the architecture in Section 4.4.

**Weaknesses:**

1. Major architecture *design choices* are not *explained*, or *not justified* by the empirical evaluation. For example:

   a) There are *three* distinct time encodings: in equation (2), in equation (9) and in the fourier features (e.g. line 252).

   b) Sharing information between sequences via the Dynamic ODE Hub is surely possible, but not an *immediate*, nor *intuitive choice*.

   c) Choosing *linear neural ODEs* for each patch, instead of a more general, neural network based neural ODE.

2. Empirical evaluation in Section 4. is seriously lacking. For example:

   a) No evaluation on *real-world data*, just synthetic data from systems with real-world relevance.

   b) No *details* about the complete train data sizes, dimensionality, or the training of baseline models LatentODE and ContiFormer.

   c) No comparison against models trained on each system *individually*. An important baseline to evaluate the zero-shot model results against.

3. An *important reference* (and probably baseline) is missing:

    "*Amortized Reparametrization: Efficient and Scalable Variational Inference for Latent SDEs*" (Course, Nair; NeurIPS 2023)

   Although presented for SDEs, the work is readily adapted to ODEs. It considers the piece-wise linear latent differential equations and mentions the idea of Transformer-based connection of the patches. Therefore, this work is very similar to the proposed Tokenized ODE Representation and intra-system attention and should be recognized as such.

4. The authors do not *motivate their problem setup* (line 093) of pure ODE inference enough. Reframing (and training) the model as a general purpose, neural ODE based time series forecasting model should be considered.

**Questions:**

1. How do you train LatentODE and ContiFormer? What architecture choices did you make for these two baselines and how many parameters do they have in total?
2. How do you evaluate all baseline models? Do they receive the same inputs as your model, i.e. all available sequences?
3. Why does your method require three distinct time series embeddings?
4. What is the intuition behind the rather complex information sharing by inducing points, including additional gating functions and GRU updates? Surely, a simple MHA (without inducing points) to share information between sequences should be a much simpler idea.
5. Do you explicitly train the extrapolation, and if so, how? By the remark on line 240, you highlight the sequence-to-sequence structure, yet you seem to be able to extrapolate 70% during evaluation (Section 4.3), just based on the last observed patch?

---

### Official Review · Reviewer_EVL8 · 2025-10-31

**Soundness:** 2
**Presentation:** 2
**Contribution:** 3
**Rating:** 4
**Confidence:** 3

**Summary:**

This paper introduces LASS-ODE, a framework that bridges large-scale foundation models with continuous-time dynamical systems. The core innovation lies in the Tokenized ODE Representation (TOR), which decomposes continuous trajectories into piecewise linear latent ODE segments, preserving continuity and interpretability while reducing integration costs. Building upon this representation, the authors design a hybrid attention architecture combining intra-system self-attention and inter-system cross-attention via a Dynamic ODE Hub. Extensive experiments demonstrate that LASS-ODE outperforms existing time-series foundation models and Neural ODE-based methods in interpolation, extrapolation and zero-shot generalization across diverse ODE systems, including chaotic systems, biological networks, and power systems.

**Strengths:**

1. The paper proposes a Large-Scale Small philosophy that first scales down ODE dynamics into compact physics-aware tokens, then scales up via global attention for system-level integration. This design effectively unifies continuous-time modeling with foundation-style scalability.

2. The combination of intra-system and inter-system attention modules with a dynamically updated ODE Hub is well-motivated and clearly described. The alternating ISA–ESA–Update mechanism effectively fuses local and global dynamics.

3. Experiments cover diverse dynamical regimes—from chaotic and oscillatory systems to power grid and biological processes. Quantitative results show consistent performance gains across interpolation, extrapolation, and zero-shot settings.

**Weaknesses:**

1. The paper does not provide a formal justification or error bound for the linear-token approximation used in TOR. A theoretical discussion on representational fidelity or stability would strengthen the contribution.

2. Although TOR aims to reduce computational costs, the model still contains ~18 M parameters (Table 3, lines 740–755). There is no direct comparison of runtime or GPU memory usage against Latent ODE or ContiFormer, which limits claims of scalability.

3. Figure 3 shows qualitative ablation results but lacks numerical values, variance, and dataset size, making it hard to assess statistical significance.

4. Several datasets are custom-simulated or combined from heterogeneous sources. The paper does not specify normalization protocols, which may hinder reproducibility.

5. While the related work discusses Physics-Informed Neural Networks (PINNs) and Hamiltonian NNs, these baselines are not included in experiments, weakening the empirical evaluation of “physics fidelity.”

**Questions:**

1. Can the authors provide an analysis or upper bound for the error induced by piecewise linearization in TOR compared to the underlying nonlinear dynamics?

2. What is the actual computational gain (in FLOPs or wall-clock time) compared with traditional Neural ODE or ContiFormer models?

3. How sensitive is LASS-ODE to the choice of token length or the number of tokens per horizon?

4. Could TOR or DOH be extended to handle non-autonomous or controlled systems as mentioned in the conclusion?

---

### Official Review · Reviewer_VNEo · 2025-11-01

**Soundness:** 3
**Presentation:** 3
**Contribution:** 3
**Rating:** 6
**Confidence:** 3

**Summary:**

The paper presented LASS-ODE (Large-Scale Small ODE), an attention-based model designed for continuous ODE trajectories across multiple dynamical systems. LASS-ODE employees two types of attentions: intra-system attention for correlations within each system and inter-system cross attention for generalizability across systems.
However, these attention mechanisms operate over discrete tokens, making it difficult to model the underlying continuous dynamics in ODEs. To address this limitation, the authors propose the Tokenized ODE Representation (TOR), a linear latent neural ODEs, that encodes local, small-scale, continuous physics inside each token. The model targets four downstream tasks: interpolation, extrapolation, probabilistic inference, and zero-shot generation across ODE systems.

**Strengths:**

1. The proposed multiscale architecture, which integrates piecewise linear latent neural ODEs for local features with attention for global features, is innovative. This design effectively bridges discrete attention mechanisms with continuous-time system representations, enabling the model to capture both local physical evolution and global structural dependencies across systems.
2. The approach is empirically assessed on a diverse set of ODE systems.

**Weaknesses:**

1. While linear neural ODEs possess great properties, they typically face challenges when modeling multicomponent stiff nonlinear systems. Accurately representing such systems using TOR may require a significantly larger number of tokens with correspondingly smaller time segments per token. The paper would benefit from a discussion on how the proposed framework handle those scenarios and what strategies might mitigate the potential limitations. Additionally, it would be valuable to analyze how properties of the dynamics parameters A and b influence the stability and expressiveness.
2. The experiment session could be more comprehensive. Notably, the evaluation does not include the probabilistic inference task mentioned in the problem setup. Furthermore, in the zero-shot generalization experiments, although LASS-ODE achieves lower MSE values compared to other methods, the absolute values remain high, raising concerns about the practical significance of the results. Additional clarifications on these aspects would strengthen the claims.

**Questions:**

1. It would be helpful to clarify whether any stability analysis was performed for the token-specific latent ODE systems. In particular, were stability constraints considered for the latent dynamics parameters, or is stability expected to be achieved purely through learning?
2. Could the authors comment on whether the seq-to-seq token reconstruction imposes a limit on the extrapolation horizon? It would also be useful to understand whether any studies were conducted on the time segments sizes for each token, especially the ones spanning beyond observation points.

---

### Official Review · Reviewer_76yg · 2025-11-06

**Soundness:** 2
**Presentation:** 2
**Contribution:** 3
**Rating:** 4
**Confidence:** 3

**Summary:**

The paper proposes a latent ODE foundation model. The main ideas are to encode the systems into global embeddings, the timepoints into localised embeddings, piecewise linearise the latent trajectories, and incorporate attention between timepoints and systems.

**Strengths:**

The results of the paper are almost magical: the performance is dramatically better than competing methods, and the forecasting visualisations show almost magical fits. I don't really understand why the method works so well, but the results are impressive nevertheless.

The main ideas of the paper are sensible and put together well.

**Weaknesses:**

The clarity of the work is ok, but could be improved. The relationships between the attentions and different embeddings could be clearer, and the fig1 is not particularly helpful.

The method is a black box, and it's difficult to see what the method learns, or why. Clearly the attention is the key part, but the paper doesn't give any insights into what the attention has learnt.

The method also feels a bit adhoc: stuff happens but not all of it is well motivated. For instance, the fourier encodings are thrown into the mix for little reason, and the ablations seem to say that they do nothing afterall.

**Questions:**

See above

---

### Note · Authors · 2025-11-15

I have read and agree with the venue's withdrawal policy on behalf of myself and my co-authors.